

# Effects on developmental outcomes after cesarean birth versus vaginal birth in Chinese children aged 1–59 months: a cross-sectional community-based survey

Hong Zhou[1,2], Yuan Ding[1], Yuning Yang[3], Siyu Zou[1], Xueqi Qu[1], Anqi Wang[1,4], Xi Wang[5], Yue Huang[1], Xintong Li[6], Xiaona Huang[3] and Yan Wang[1,2]

[1] Department of Maternal and Child Health, School of Public Health, Peking University, Beijing, China
[2] Environmental and Spatial Epidemiology Research Center, National Human Genetic Resources Center, Beijing, China
[3] UNICEF China, Beijing, China
[4] Department of Preventive Medicine, Keck School of Medicine, University of Southern California, Los Angeles, CA, United States of America
[5] Children's Hospital of Philadelphia, Philadelphia, PA, United States of America
[6] Beijing Camford Royal School, Beijing, China

Corresponding author
Xiaona Huang, xhuang@unicef.org

## ABSTRACT

**Objective**. It is controversial whether the mode of delivery is associated with developmental outcome, and little was known about growth and development of cesarean children in poor rural areas in China. We aim to measure the development of both cesarean and vaginal-delivered children by Ages and Stages Questionnaires (ASQ) and explore the association between mode of delivery and developmental outcome in poor rural areas in China.

**Methods**. Data were collected from a cross-sectional community-based survey, which recruited 1,755 vaginal delivered and cesarean children ages 1 to 59 months in eight counties of China. Caregivers of those children completed the Chinese version of ASQ-3 (ASQ-C) while physical examination andquestionnaires on socio-demographic and neonatal characteristics were conducted. Multivariate logistic regressions were used to measure the association between developmental delay and mode of delivery as well as each socio-demographic factor, respectively, after adjusting other socio-demographic characteristics.

**Results**. The prevalence of suspected overall developmental delay was 23.4% in the cesarean group, compared with 21.3% in the vaginal delivered group, yet without statistical difference ($p < 0.05$). Developmental delay was also not significantly different between cesarean and vaginal delivered group in five ASQ domains of communication (7.7% vs. 7.8%, $p = 0.949$), fine motor (7.0% vs. 6.1%, $p = 0.538$), gross motor (8.5% vs. 6.4%, $p = 0.154$), problem solving (7.2% vs. 6.7%, $p = 0.722$) and personal social (8.0% vs. 7.9%, $p = 0.960$).

**Conclusions**. Our findings suggest that cesarean delivery does not increase or decrease the risk of suspected developmental in children delay as compared with vaginal delivery.

## INTRODUCTION

According to published data from 150 countries in 1990–2014 (*Betran et al., 2016*), the global cesarean section rate rose to 18.6%, which was higher than the 15% recommended cesarean section rate proposed by WHO in 1985 (*1985*). With the change of time, cesarean section rate has risen rapidly in both developed and developing countries, and the worldwide increasing trend caused many concern (*Betran et al., 2016*; *Vogel et al., 2015*).

With the development of perinatal care, the safety of cesarean section has been greatly improved, and cesarean section plays an important role in reducing maternal and neonatal mortality. When cesarean section rate rose from less than 5% to 10%, the cesarean section rate was significantly associated with the decreased maternal and neonatal mortality, especially in low-income countries where cesarean section was not available (*Althabe et al., 2006*). However, WHO pointed out that caesarean section can effectively reduce the maternal and neonatal death only when it is medically justified, and there is no evidence that caesarean delivery can provide benefit to the mother and child who do not require the procedure in 2015 (*Betran et al., 2016*; *Hannah et al., 2000*). At population level, once cesarean section rate reached 10%, its further increased rate is not associated with a reduction of maternal, neonatal, and infant mortality rates (*Ye et al., 2014*).

There were some mechanisms explaining how the process of cesarean section may affect children's neurodevelopment. First, cesarean children lack the normal extrusion process of birth canal. The birth canal extrusion during vaginal delivery can effectively help with fetal lung expansion. In the delivery, the baby's head and chest are constantly squeezed, stimulating the respiratory center and removing excess fluid from the lung, which help the baby to breathe smoothly after birth (*Hooper & Harding, 1995*; *Milner, Saunders & Hopkin, 1978*; *Vyas, Milner & Hopkins, 1981*). The sympatho-adrenal activity in the fetal body also increased significantly to face against the pressure (*Falconer & Poyser, 1986*), which not only help the baby face the whole delivery process, but also make them better adapted to the outer environment and promote their neurobehavioral development. Second, instrumental and surgical delivery was associated with compromised early mother-infant interaction and a higher risk of breastfeeding failure (*Rowe-Murray & Fisher, 2001*). The postnatal recovery for vaginal delivery is faster, only taking 1–3 days. Women can eat immediately after childbirth and get out of bed the day or the following day after the delivery of a baby. In contrast, it usually takes 5–7 days for women to get recovered after cesarean section (*Donowitz & Wenzel, 1980*; *Rortveit et al., 2003*). Therefore, women after vaginal delivery have more time and energy to take care of the newborn and establish a good maternal-child interaction, which may benefit neonatal neurological development (*Robson et al., 2015*; *Rowe-Murray & Fisher, 2001*). In addition, women after vaginal delivery have fewer complications (*Arikan et al., 2012*) and can begin breastfeeding sooner than cesarean born infants. The early interaction and sucking behavior are of great importance in establishing successful breastfeeding (*Sakalidis et al., 2013*). Women who had cesarean section had a higher proportion of breastfeeding difficulties and a less breastfeeding rate (*Hobbs et al., 2016*; *Prior et al., 2012*). Breastfeeding can strengthen the maternal-child interaction and attachment, which plays an important role in young children's emotional development.

However, the existing research evidence is not consistent relating to the effects of cesarean section on child health outcomes. For example, some studies have shown that the cesarean section was not associated with a reduction in risk of death or neurodevelopmental delay in children (*Asztalos et al., 2016*; *Bahl et al., 2007*; *Dekeunink et al., 2016*; *Haque et al., 2008*; *Joseph et al., 2015*; *Kimura et al., 2017*; *Robson et al., 2015*; *Spinillo et al., 1992*; *Whyte et al., 2004*; *Zhu et al., 2014*), while others argued that cesarean section significantly reduced the risk of children neurodevelopmental delay in a specific case (*Molkenboer et al., 2006*). Some other studies showed that cesarean section may have a negative impact on the baby's respiratory system and immune system (*Lee et al., 2014*; *Shearer, 1993*). According to the statement of the WHO, the impact of cesarean section on maternal and child health still remains unclear, and more research is needed to understand the short- and long-term effects of cesarean section on children's health outcomes.

As China's cesarean section rate was up to 46% according to the WHO survey in 2008, and up to date, it has remained at the highest level in the world (*Lumbiganon et al., 2010*), this has provided a good opportunity for the study of effects of cesarean section on child health outcomes. According to the latest research covering 2865 counties in mainland China's 31 provinces, China's overall cesarean section rate is 34.9%, with the rate declining in some of the largest cities and still rising up in poor rural areas (*Lee et al., 2014*). Moreover, China's rural areas have a large birth population, it is imperative to understand the impact of high cesarean section level on children's physical, psychological and intellectual development.

## Objective

We aim to measure the development of both cesarean and vaginal-delivered children by Ages and Stages Questionnaires (ASQ) and explore the association between mode of delivery and developmental outcome in poor rural areas in China.

## Design

Our study was a cross-sectional community-based survey on early child development as part of maternal and child health program funded by UNICEF in 2016. The study covered 8 rural counties of four provinces in China (Xinjiang, Qinghai, Jiangxi and Ningxia), with the total population of 3,639,000 in the year of 2016. The annual income per capita of the project counties was 8997 RMB, lower than the national average of 12,363 RMB in rural areas in 2016.

### Setting and subjects

Multistage sampling method was used to select the townships and villages from each county. First, 15 administrative villages per county and then two nature villages per administrative village were selected at random with population proportional to size (PPS). Second, within each selected nature village, 10 households with a child under 3 years of age were selected by simple random sampling approach according to the children registration list provided by local village doctors. All the sampling processes were completed by the staff members of the program and random numbers for selection criteria were generated by a random number table. The study was approved by Ethical Committee of Peking University Health Science

Centre (IRB00001052-16041). The study procedures were explained to all the caregivers involved in the study and their written informed consent to the study was obtained prior to their participation in the study.

In our study, the statistical significance ($\alpha$) was set at 0.05 level and the power of the test (1- $\beta$) was 80%. As the study was designed to collect relevant baseline data for an intervention project, we assumed the baseline prevalence of suspected development delay (p0) would be 40% and expected a relative 20% decrement of prevalence (p1 –p0) / p0. Finally, in consideration of an 80% response rate, the needed total sample size of children would be 1354 (677:677). The following formula was used to calculate the sample size:

$$n = 2\overline{pq}(Z_\alpha + Z_\beta)^2/(p_1 - p_0)^2, \overline{p} = (p_1 + p_0)/2, \overline{q} = 1 - \overline{p}.$$

## Main outcome measures

Chinese version of the Ages & Stages Questionnaires, Third Edition (ASQ-C) was used to evaluate children's neurodevelopment in our study (*Bian et al., 2012*; *Wei et al., 2015*). The ASQ-C is a series of 21 parent/caregiver-completed questionnaires designed as an alternative screening assessment of developmental performance among children aged 1–66 months, and the difficulty of the questions increases with the increasing age of the children. The ASQ-C contains five major developmental domains (communication, gross motor, fine motor, problem solving and personal-social skills), and each domain consists six questions. A caregiver was asked about her/his child's behaviour, such as ''Does your baby pick up a crumb or Cheerio with the tips of his thumb and a finger''. The response to each question is one of the following: ''yes'', ''sometimes'' or ''not yet'' and scores are 10, 5, 0 points respectively. The score on each of the six questions was summed to obtain an ASQ-C domain score. We considered a certain domain score as abnormal if it was less than two standard deviations from the mean of the Chinese reference group. In our study, the ASQ result is considered to be abnormal if the score for any 1 of the 5 domains is abnormal (*Squires, Potter & Bricker, 1995*). The abnormal ASQ result indicated a suspected developmental delay of the child. The assessment process took approximately 10–15 min.

Information on child's age, gender (boys and girls), birth order (1, 2, 3, or more), birthweight, gestational age at birth and delivery method (cesarean section or vaginal delivery) was included in our study, and information on caregiver's education (illiterate, primary school, secondary school, college, and above) and income (poorest, poor, middle, richer, and richest) was also collected. All data collection was accomplished by tablet computers with the Good Data entry system. Our questionnaire was adjusted based on a pilot survey conducted in the Yulong County in Yunnan Province. Before the formal fieldwork was conducted, all the interviewers involved were trained by the study guideline, including the procedure to use the digital survey equipment, to conduct a household survey, and to understand the standards and requirements of ASQ assessments.

## Statistical analysis

All children involved in our study were full-term delivered single children, with 37–42 weeks of gestation. They were divided into two groups according to the different ways of delivery: cesarean and vaginal-delivered

**Table 1** Characteristics of children aged 1–59 months with cesarean section and vaginal delivery groups in eight counties in China (2016).

| Characteristics | Cesarean section | Vaginal delivery | P |
|---|---|---|---|
| **Total, *n* (%)** | 401(22.8) | 1,354 (77.2) | |
| **Gestational age weeks, mean (range)** | 39+0(37+0-41+6) | 39+2(37+0-41+6) | <0.001 |
| **Birthweight g, mean (range)** | 3,198 (1,500–4,600) | 3,187 (1,700–5,000) | 0.757 |
| **Age of child, months, *n* (%)** | | | 0.601 |
| 1–11 | 83(20.7) | 255(18.8) | |
| 12–23 | 99(24.7) | 361(26.7) | |
| 24–59 | 219(54.6) | 738(54.5) | |
| **Gender of child, *n* (%)** | | | 0.467 |
| Boys | 210(52.4) | 737(54.4) | |
| Girls | 191(47.6) | 617(45.6) | |
| **Birth order of child, *n* (%)** | | | 0.051 |
| 1 | 175(43.6) | 519(38.3) | |
| 2 | 169(42.1) | 580(42.8) | |
| ≥3 | 57(14.2) | 255(18.8) | |
| **Caregiver's education, *n* (%)** | | | <0.001 |
| Illiteracy | 46(11.5) | 181(13.4) | |
| Primary | 80(20.0) | 334(24.7) | |
| Secondary | 224(55.9) | 757(55.9) | |
| College and above | 51(12.7) | 82(6.1) | |
| **Low birthweight, *n* (%)** | | | 0.219 |
| Yes | 24(6.1) | 60(4.6) | |
| No | 368(93.9) | 1,248(95.4) | |
| **Income, *n* (%)** | | | 0.080 |
| Poorest | 93(24.1) | 289(22.0) | |
| Poor | 50(13.0) | 246(18.7) | |
| Middle | 85(22.0) | 283(21.6) | |
| Richer | 58(15.0) | 159(12.1) | |
| Richest | 100(25.9) | 336(25.6) | |

The prevalence of suspected developmental delay in cesarean and vaginal-delivered children were calculated and compared using chi-square tests. Then the multivariate logistic regression analyses were used to examine the relationship between all the socio-demographic factors (especially the method of delivery) and the suspected developmental delay, leading to crude, adjusted odds ratios, and 95% confidence interval for the suspected developmental delay. The socio-demographic factors included the child's age, gender, birth order, birthweight, delivery method and the caregiver's education and income. All analyses were conducted using SPSS version 25.0.

## RESULTS

As shown in Table 1, a total of 1,755 children aged 1–59 months were included in our study. Among 1,755 children, 401 (22.8%) of them were delivered by cesarean section.

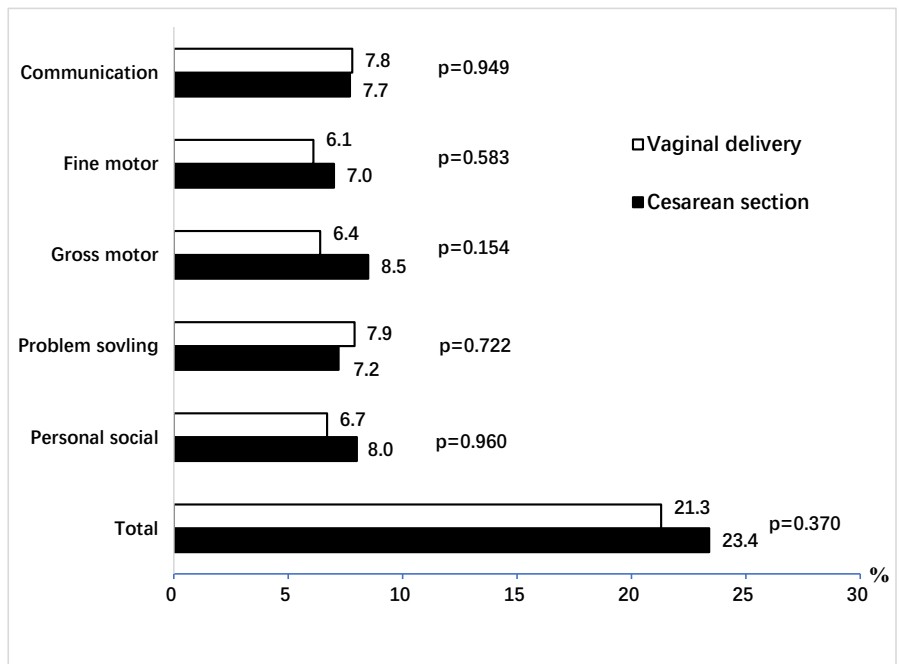

**Figure 1** **Comparison of prevalence of suspected developmental delay in cesarean section and vaginal delivery groups in eight counties of China, 2016.** There was difference in the prevalence of overall suspected developmental delay between cesarean group (23.4%) and vaginal delivered group (21.3%), but this difference was not statistical significant ($p = 0.37$). Meanwhile, the rate of suspected developmental delay in cesarean group for communication, fine motor, gross motor, problem solving and personal social was 7.7%, 7.0%, 8.5%, 7.2% and 8.0%, respectively, and there was no significant difference in these five separate ASQ domains between two groups.

There were significant differences in gestational age weeks ($p < 0.001$) and the caregiver's education ($p < 0.001$) between cesarean and vaginal delivered children (Table 1).

The prevalence of suspected developmental delay in cesarean and vaginal delivered groups was similar (23.4% vs 21.3%; $p = 0.37$) (Fig. 1). There was no significant difference in the five separate ASQ domains between the two groups (Fig. 1).

Table 2 showed the bivariate and multivariate regression output of risk factors associated with suspected developmental delay among children in 8 counties in China. The variations of abnormal ASQ scores significantly differed by household income. Children from the poorest quintile families tended to have the poorest performance on ASQ, with abnormal rate of 27.0% and adjusted OR of 1.48 (95% CI [1.04–2.11], $p < 0.05$). Boys had a higher proportion of abnormal ASQ scores in comparison to girls (24.5% vs. 18.7%, adjusted OR=1.31, 95% CI [1.03–1.67]). The cesarean group had a higher proportion of abnormal ASQ scores as compared to vaginal delivery group (23.4% vs. 21.3%, adjusted OR = 1.24, 95% CI [0.94–1.64]), but the difference is not significant.

**Table 2  Socio-demographic factors for abnormal ASQ scores among children in eight counties in China (2016).**

| Characteristics | N | n | %(95%CI) | Crude OR(95% CI) | Adjusted OR(95% CI)[a] |
|---|---|---|---|---|---|
| **Total** | 1,755 | 383 | 21.8(19.9–23.8) | | |
| **Age of child, months** | | | | | |
| 1–11 | 338 | 64 | 18.9(14.8–23.1) | 1 | 1 |
| 12–23 | 460 | 113 | 24.6(20.6–28.5) | 1.39(0.98–1.96) | 1.39(0.97–2.00) |
| 24–59 | 957 | 206 | 21.5(18.9–24.1) | 1.17(0.85–1.60) | 1.17(0.85–1.63) |
| **Gender of Child** | | | | | |
| Boys | 947 | 232 | 24.5(21.8–27.2) | 1.41(1.12–1.77)[*] | 1.31(1.03–1.67)[*] |
| Girls | 808 | 151 | 18.7(16.0–21.4) | 1 | 1 |
| **Birth order of child** | | | | | |
| 1 | 694 | 132 | 19.0(16.1–21.9) | 1 | 1 |
| 2 | 749 | 170 | 22.7(19.7–25.7) | 1.25(0.96–1.61) | 1.14(0.87–1.51) |
| ≥3 | 312 | 81 | 26.0(21.1–30.8) | 1.49(1.08–2.04)[*] | 1.31(0.93–1.83) |
| **Caregiver's education** | | | | | |
| Illiteracy | 227 | 45 | 19.9(14.6–25.0) | 1.68(0.92–3.08) | 1.45(0.73–2.90) |
| Primary | 414 | 109 | 27.5(22.1–30.6) | 2.43(1.40–4.24)[*] | 2.28(1.20–4.32)[*] |
| Secondary | 981 | 212 | 19.6(19.0–24.2) | 1.88(1.10–3.19)[*] | 1.88(1.02–3.46)[*] |
| College and above | 133 | 17 | 15.9(7.1–18.5) | 1 | 1 |
| **Income** | | | | | |
| Poorest | 382 | 103 | 27.0(22.5–31.4) | 1.66(1.19–2.32)[*] | 1.48(1.04–2.11)[*] |
| Poor | 296 | 63 | 21.3(16.6–25.9) | 1.22(0.84–1.76) | 1.13(0.77–1.67) |
| Middle | 368 | 81 | 22.0(17.8–26.2) | 1.27(0.90–1.80) | 1.21(0.84–1.75) |
| Richer | 217 | 50 | 23.0(17.4–28.6) | 1.35(0.90–2.01) | 1.46(0.96–2.21) |
| Richest | 436 | 79 | 18.1(14.5–21.7) | 1 | 1 |
| **Low birthweight, n (%)** | | | | | |
| Yes | 84 | 22 | 26.2(16.8–35.6) | 1.31(0.79–2.16) | 1.37(0.82–2.29) |
| No | 1,616 | 344 | 21.3(19.3–23.3) | 1 | 1 |
| **Mode of delivery, n (%)** | | | | | |
| Cesarean section | 401 | 94 | 23.4(19.3–27.6) | 1.12(0.86–1.47) | 1.24(0.94–1.64) |
| Vaginal delivery | 1,354 | 289 | 21.3(19.2–23.5) | 1 | 1 |

**Notes.**
[a] Adjusted for all the variables in Table 2.
[*] $p < 0.05$.

# DISCUSSION

Our study shows that there is no significant difference in neurodevelopmental behaviors between cesarean and vaginal-delivered children. The abnormal ASQ rate was a reflection of developmental delay in children, but we did not find any significant difference in general result or each ASQ domain between the two groups of children.

Our results are consistent with other studies regarding full-term, preterm, breech presentation children or twins, and these studies also found no significant difference in developmental and behavioral outcomes between cesarean and vaginal-delivered groups (*Asztalos et al., 2016*; *Bahl et al., 2007*; *Dekeunink et al., 2016*; *Haque et al., 2008*;

*Joseph et al., 2015*; *Kimura et al., 2017*; *Robson et al., 2015*; *Spinillo et al., 1992*; *Zhu et al., 2014*). However, previous studies have found some negative outcomes due to cesarean delivery. For example, Cesarean delivery was related to an increased risk of attention-deficit/hyperactivity disorder or autism spectrum disorder (*Curran et al., 2015a*; *Curran et al., 2015b*; *Talge, Allswede & Holzman, 2016*). Studies in the UK and Iceland found that birth by caesarean section was associated with an increased risk of attention-deficit/hyperactivity disorder (ADHD) or autism spectrum disorder (ASD) compared with spontaneous vaginal delivery, especially birth by emergency cesarean section, or by cesarean section plus induction of labor (*Curran et al., 2015a*; *Curran et al., 2016*; *Valdimarsdottir et al., 2006*). Cesarean children also reported more hearing screening failure than vaginal delivered children (*Smolkin et al., 2013*; *Xiao et al., 2015*).

Notably, the rate of cesarean section in our study area was 22.8%, lower than the national average rate (*Li et al., 2017*), but still above the 10–15% recommended by WHO. The same trend was also shown in other economic developed and developing counties in China. Reasons for the increased rate of cesarean delivery were complicated, and many women chose cesarean delivery without medical indications because of their fear of pain and the belief that cesarean delivery was more beneficial for both mother and baby than vaginal delivery (*Huang et al., 2012*). Another reason is the affordability of the cost of cesarean section in China. China's new cooperative medical scheme (NCMS) was launched in 2003 to provide families health insurance in rural areas so that more families in rural areas could afford to have a cesarean section (*Huang et al., 2012*). This result is consistent with the studies in USA, Italy and Brazil that the rate of caesarean section in private hospitals was significantly higher than in public hospitals, possibly because women who had private insurance would like to choose cesarean sections, many of which had no medical indications (*Giani et al., 2011*; *Hopkins, De Lima Amaral & Mourao, 2014*; *Lipkind et al., 2009*). Cesarean sections not only brought financial burden to patients and their families especially in poor areas (*Bogg et al., 2010*; *Deboutte et al., 2015*), but also placed an economic burden on already highly stressed medical systems worldwide (*Druzin & El-Sayed, 2006*).

The World Health Orgaization recommends that cesarean section should be performed according to medical indications to minimize unnecessary cesarean section. It is still unclear whether all the caesarean sections conducted in the surveyed area were necessary. Further studies were needed to discover the appropriate rate of cesarean section for Chinese rural areas. The Robson classification system, as a global standard for assessing and monitoring caesarean sections (*Betran et al., 2016*), is not widely adopted in most poor rural areas of China. We need to establish a reliable and internationally recognized classification system especially in Chinese rural areas, which can be further compared with the international results.

We also noticed other sociodemographic factors were associated with developmental outcomes. In our study, boys are at higher risk of developmental delay, and similar results were also found in Dutch, Norway, UK and Egypt among children aged 4-60 months, showing a significantly higher average ASQ score of girls especially in communication, personal social and fine motor domains (*Abo El Elella et al., 2017*; *Kerstjens et al., 2009*;

*Morley et al., 2015*; *Richter & Janson, 2007*). Low income is another high risk factor affecting the development of young children, which was also reflected in previous studies (*Handal et al., 2007*; *Noble et al., 2015*; *Seguin et al., 2005*).

There are a number of limitations in the present study. First, only ASQ-C screening scale, was used to evaluate the developmental status of children. Children who have developmental delay cannot be firmly diagnosed. Second, a cross-sectional study was used that whether cesarean section had an impact on children's long-term behavior or psychological development was not determined. A future study using a longitudinal study design is needed to investigate the effects of cesarean section on child health outcomes. Third, insufficient information was collected to assess if the cesarean deliveries were in line with medical indications, so we were not able to distinguish the impact of necessary and unnecessary cesarean section on the child developmental outcomes.

In conclusion, our study showed the prevalence of suspected developmental delay in cesarean and vaginal delivered were similar in poor rural areas in China. Therefore, our findings suggest that cesarean delivery does not increase or decrease the risk of suspected developmental in children delay as compared with vaginal delivery.

## ACKNOWLEDGEMENTS

We would like to thank research teams from Lanzhou University and Capital Medical University for their hard work in orchestrating the field work and identifying study population. We also want to thank all of family members who participated in this study from the eight counties in rural China.

### Funding

This work was funded by a grant from UNICEF China (YH702). The funders had no role in study design, data collection and analysis, decision to publish, or preparation of the manuscript.

### Grant Disclosures

The following grant information was disclosed by the authors:
UNICEF China: YH702.

### Competing Interests

Yuning Yang and Xiaona Huang are employed by UNICEF China.

### Author Contributions

- Hong Zhou conceived and designed the experiments, performed the experiments, contributed reagents/materials/analysis tools, prepared figures and/or tables, authored or reviewed drafts of the paper, approved the final draft.
- Yuan Ding analyzed the data, contributed reagents/materials/analysis tools, prepared figures and/or tables, approved the final draft, literature review and polished the language.

- Yuning Yang performed the experiments, prepared figures and/or tables, approved the final draft.
- Siyu Zou analyzed the data, prepared figures and/or tables, approved the final draft.
- Xueqi Qu approved the final draft, literature review and polished the language.
- Anqi Wang prepared figures and/or tables, approved the final draft, literature review and polished the language.
- Xi Wang approved the final draft, literature review and polished the language.
- Yue Huang analyzed the data, contributed reagents/materials/analysis tools, prepared figures and/or tables, approved the final draft.
- Xintong Li prepared figures and/or tables, approved the final draft, literature review and polished the language.
- Xiaona Huang and Yan Wang performed the experiments, prepared figures and/or tables, approved the final draft.

## Human Ethics

The following information was supplied relating to ethical approvals (i.e., approving body and any reference numbers):

Ethical Committee of Peking University Health Science Centre approved this study (IRB00001052-16041).

## Data Availability

The raw measurements are available in the Supplemental File.

## Supplemental Information

Supplemental information for this article can be found online at http://dx.doi.org/10.7717/peerj.7902#supplemental-information.

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
