# Peer review of "Effects on developmental outcomes after cesarean birth versus vaginal birth in Chinese children aged 1–59 months: a cross-sectional community-based survey"

_PeerJ, doi:10.7717/peerj.7902_

## Round 0.1 · original submission · Major Revisions

The manuscript was reviewed by two experts n the field. They raised substantial comments that need to be addressed before this paper can be considered for publication

Reviewer 1 ·

Basic reporting

There are a few sentences that need rephrasing for clarity.

Introduction. I suggest that you improve the description of the objective at lines 112-115.

Consider the objective of the abstract.

Although your results are compelling, the interpretation should be improved in the following ways:

Table 1. Among 1755 children, 102(22.8%) of them were delivered by cesarean section.

According to Table 1, they are 401 (22.8%).

The results must be presented in text or table, but not both. Your results are repetitive.

Lines 182-186 and 188-190 are already included in Table 1, I suggest deleting those paragraphs.

I suggest to the authors should include all results relevant to the hypothesis (significant p-values). For example, “A total of 1755 children aged 1-59 months were included in our study, 401 (22.8%) of them were delivered by cesarean section. There were significant differences in gestational age weeks (p<0.001) and the caregiver’s education (p<0.001) between cesarean and vaginal delivered children (Table 1)”.

Figure 1. Lines 194-197 are already included in Figure 1, I suggest deleting those paragraphs. Consider "The prevalence of suspected developmental delay in cesarean and vaginal delivered groups was similar (23.4 % vs 21.3%; p=0.37) (Figure 1). There was no significant difference in the five separate ASQ domains between the two groups (Figure 1)."

The p-values that were included in the Abstract are not included in the results section. I suggest to authors include those p-values in Figure 1 "ASQ domains of communication (7.7% vs. 7.8%, p=0.949), fine motor (7.0% vs. 6.1%, p=0.538), gross motor (8.5% vs. 6.4%, p=0.154), problem solving (7.2% vs. 6.7%, p=0.722) and personal social (8.0% vs. 7.9%, p=0.960)."

In Tables 2 and 3, I suggest deleting following sentences: “Among these children, 21.3% of them had abnormal ASQ scores” and “Among these children, 23.4% of them had abnormal ASQ scores.

Experimental design

The main limitation of this study is the design. A cross-sectional study is not the most appropriate to measure the development of both cesarean and vaginal-delivered children. The design should be a cohort study (Asztalos et al. Twin Birth Study: 2-year neurodevelopmental follow-up of the randomized trial of planned cesarean or planned vaginal delivery for twin pregnancy. Am J Obstet Gynecol. 2016;214(3):371.e1-371.e19). However, the authors describe this in the study limitation and conclusion section. Despite this limitation, the methods “cross-sectional study” were described with sufficient detail and information to replicate.

Validity of the findings

The conclusions should be appropriately stated and should be limited to those supported by the results. However, in this study, the conclusions are not well stated. These are not limited to supporting results.

If the prevalence of suspected developmental delay in cesarean and vaginal delivered groups was similar (23.4 % vs 21.3%; p=0.37), Why do they conclude that cesarean birth had no effects on children’s growth and development?

I suggest that you improve the description of conclusion, for example, “Our study showed the prevalence of suspected developmental delay in cesarean and vaginal delivered were similar in poor rural areas in China. Therefore, our findings suggest that cesarean delivery does not increase or decrease the risk of suspected developmental in children delay as compared with vaginal delivery”

Additional comments

I suggest you include the study design in the descriptive title, for example, "Effects on developmental outcomes after cesarean birth versus vaginal birth in Chinese children aged 1-59 months: A cross-sectional community-based survey"

Consider in the manuscript the Methods and Discussion sections.

Reviewer 2 ·

Basic reporting

Is an interesting and well writing paper, with a huge number of patients.
In a cross sectional design, with the limitations discussed by the authors, the aim of the study is to compare the prevalence of developmental delay risk in cesarean and vaginal born children, and to constructo a multivariate model for both groups.

If there is no differences in the prevalence of developmental delay (DD) risk between cesarean and vaginal born children, no interaction is expected between DD risk and way of delivery, for this reason I think that authors can construct a global multivariate model, instead of different model for each group.

In figure 1: please add a foot note explaining better the figure and analysis. ASQ has 5 domain, nos 6. In the figure there is one column called "stunting", is not clear what does it means, neither this concept is explained in methodology section.

The paper lack of references about ASQ validation in China. Please add them.

Experimental design

Is not an experimental review

Validity of the findings

I agree with the validity of the finding

---

## Round 0.2 · Minor Revisions

The manuscript received favorable reviews

However please address the remaining comments of reviewer number 2 that need to addressed before this manuscript can be considered for publication

Reviewer 1 ·

Basic reporting

I have no commens.

Experimental design

I have no comments.

Validity of the findings

I have no comments.

Additional comments

The authors have corrected the comments. I suggest accept the manuscript.

Reviewer 2 ·

Basic reporting

I agree with the changes made by the authors.

I recommend to erase the sentence in the discussion: "Our study did not find evidence to support cesarean section would improve child’s growth and neurodevelopment." (second sentence of discussion, lane 212), because in the same sentence it is stated: "Our results are consistent with other studies regarding full-term, preterm, breech presentationchildren or twins, and these studies also found no significant difference in developmental and behavioral outcomes between cesarean and vaginal-delivered groups"

Please review the footnote in table 2, there are asterisks, a and b, and it is not clear the difference between b and asterisks. >=3 child birth order is not significant in the adjusted analysis.

In other hand, the recommendation to construct a multivariable analysis is including only the significant variables in the crude analysis, not all of them, please review with a statistician.

Experimental design

Adequate

Validity of the findings

Adequate

---

## Round 0.3 · accepted · Accept

The authors have addressed all the comments of the reviewers and the manuscript can be accepted for publication.